# Empowering Children as Agents of Change to Foster Resilience in Community: Implementing “Creative Health” in Primary Schools after the Fukushima Nuclear Disaster

**DOI:** 10.3390/ijerph19063417

**Published:** 2022-03-14

**Authors:** Aya Goto, Alison Lloyd Williams, Satoko Okabe, Yohei Koyama, Chihaya Koriyama, Michio Murakami, Yumiya Yui, Kenneth E. Nollet

**Affiliations:** 1Center for Integrated Science and Humanities, Fukushima Medical University, Fukushima 960-1295, Japan; ykoyama@fmu.ac.jp; 2Department of Sociology, Lancaster University, Lancaster LA1 4YW, UK; a.lloydwilliams@lancaster.ac.uk; 3Department of Food and Nutrition, Koriyama Women’s University, Fukushima 963-8503, Japan; okabe@koriyama-kgc.ac.jp; 4Department of Epidemiology and Preventive Medicine, Graduate School of Medical and Dental Sciences, Kagoshima University, Kagoshima 890-8544, Japan; fiy@m.kufm.kagoshima-u.ac.jp; 5Department of Health Risk Communication, School of Medicine, Fukushima Medical University, Fukushima 960-1295, Japan; michio@cider.osaka-u.ac.jp; 6Center for Infectious Disease Education and Research, Osaka University, Osaka 565-0871, Japan; 7Department of Public Health and Health Policy, Graduate School of Biomedical and Health Sciences, Hiroshima University, Hiroshima 734-8553, Japan; yumiya@hiroshima-u.ac.jp; 8Department of Blood Transfusion and Transplantation Immunology, Fukushima Medical University, Fukushima 960-1295, Japan; nollet@fmu.ac.jp

**Keywords:** children, Fukushima nuclear accident, humanities, arts, community networks, resilience

## Abstract

The “Creative Heath” project, a participatory school activity to foster community resilience, was implemented in Fukushima, Japan, and children’s experiences of the project were assessed both quantitatively and qualitatively. The project consists of three workshops: BODY, FOOD, and ACT, with activities to facilitate students’ scientific and creative thinking, working in teams, presenting, and expressing their opinions. The first two schools participated with 105 students aged 9–11 years old. Before and after each workshop, students were given questionnaires to rate their satisfaction with their own health (BODY), local foods (FOOD), and the community at large (ACT) on a five-level scale, with space to add free comments. Ratings for BODY and FOOD changed significantly, and the proportion of students who increased their rating of an evaluation indicator after each workshop was 25% for BODY, 28% for FOOD, and 25% for ACT. Text analysis of free comments showed that students in the “increased” group appreciated presenting, measuring, learning connections between nutrition and health, and working collaboratively with peers. Children perceived their health and the foods in their community more positively after participating. Moreover, the Creative Health project could be a way to enhance children’s creativity and autonomy as agents of change in the community.

## 1. Introduction

The Fukushima Daiichi nuclear power plant disaster, a consequence of the 2011 Great East Japan Earthquake, affected the lives of families with young children because of uncertainties related to long-term, low-dose radiation exposure. A high prevalence of maternal depression was reported in the early years [1], but on a positive note, the depression prevalence decreased with time [2] and maternal confidence was not affected by the disaster environment [3]. Similar resilience among mothers and children was reported following the Chernobyl nuclear disaster, after which Bromet worked with and followed affected mothers for 11 years and found that the disaster experience was reflected in how they perceived their children’s well-being, but their anxiety was not transmitted to the children themselves [4]. A comparison of mental health and teachers’ evaluations did not differ between the affected and matched non-affected children [4]. Bromet followed Chernobyl evacuee children even longer (19 years) and reported that negative risk perceptions of the nuclear disaster reported by teens were only modestly associated with their mental health [5]. Similar observations in Fukushima [6] suggest constructive alternatives to simply treating children as a vulnerable group while focusing on negative mental health [7] and physical consequences [8].

In the 2015–2030 Sendai Framework for Disaster Risk Reduction, it is stated that “Children and youth are agents of change and should be given the space and modalities to contribute to disaster risk reduction, in accordance with legislation, national practice and educational curricula” [9]. More studies and interventions to understand and promote roles that children can take in disaster recovery are needed in Japan. The Ministry of Education, Culture, Sports, Science, and Technology (MEXT) announced New National Curriculum Standards in 2017, to be implemented from 2020 to 2022. The new standards aim to shift school education from “what teachers have to teach” to “what students will be able to do and how can they learn” [10]. With this shift, they recommend building school-community partnerships in order to equip students with competencies and skills to live and engage in the community.

An overseas example of fostering children’s community participation in the face of a crisis is Lancaster University’s work after the United Kingdom’s 2013–2014 winter floods. Their team researched the active roles that young people can play in disaster management by using the arts [11,12]. This innovative approach was expanded to a European research project named CUIDAR, funded by the European Union’s Horizon 2020 program [13]. The CUIDAR project aimed to enhance the resilience of children and young people after disasters and enable disaster responders to meet their needs more effectively through a participatory approach. In particular, we adapted the participatory theater approach of Lloyd Williams to work with Fukushima children in the Japanese education system. The project content and participating teachers’ responses were reported previously [14]. In brief, the method, drawn from Theatre for Development [15] and Theatre of the Oppressed [16], invites children to use performance to explore what their community means to them. The approach relies on the principle that the theatre space spans the worlds of “reality” and the “image of reality” [17], thus children use their bodies and voices to physically ‘map’ their community as they see and experience it and play with alternative visions of what that community could be. The approach invites children to step in and out of scenes, reflecting critically on what they have created and asking questions about what more they might want to know or find out. Thus, the work invites children to take the lead in identifying issues of importance in their local community.

Based on the successful implementation of the participatory theater approach and inspired by a report titled “Creative Health” by the UK All-Party Parliamentary Group (APPG) on Arts, Health and Wellbeing [18], we have expanded our project by adding medical and nutritional components, developed teaching tools, and started implementing our own “Creative Health” project at local schools. The APPG used the word “arts” to go beyond its traditional view of highly intellectual products and defined it as “shorthand for everyday human creativity”. They stated that “The act of creation, and our appreciation of it, provides an individual experience that can have positive effects on our physical and mental health and wellbeing”. In the MEXT Standards, “creativity” is defined as “the quality and ability to create new meanings and values based on one’s own thoughts and ideas, while making rich use of sensitivity” [19]. Our study aims were twofold: first, to report the content of the “Creative Health” project; and second, to present quantitative and qualitative assessments of Fukushima children’s experiences in the project. We furthered the discussion of how our project matched the MEXT’s new standards.

## 2. Materials and Methods

### 2.1. Project Development

The “Creative Health” project aims to equip children with critical thinking skills and to empower them to take active roles in the community. When facing a disaster or health crisis, adults tend to regard children as a vulnerable group in need of protection, rather than as a group of active participants in a disaster or crisis response. Alternatively, adults can appreciate that children have their own knowledge and understanding of what is happening in a community, and their experiences in a disaster can contribute a great deal to the building of community resilience. This perspective aligns with the MEXT New National Curriculum Standards.

The project is built around three workshops: BODY, FOOD, and ACT (Figure 1). Initially, the first and second authors (AG and ALW) modified the UK model “participatory theater approach” [11] and developed ACT, as mentioned above. In the ACT workshop, students have the opportunity to express what they think about food and health in their local community. Next, we added FOOD and BODY workshops that were designed by the third and last authors (SO and KEN), respectively. In the BODY workshop, students start by sharing simple statements of what they know or believe about the heart, lungs, and/or blood, continue by exploring related medical discoveries and the people who made them, and then wrap up by making their own observations, e.g., of heart rate and how it varies. In the FOOD workshop, students learn how their body depends on what they eat and how their community is responsible for producing and/or procuring the foods they need. In these two workshops, we focused on the circulation of blood and the prevention of anemia as underlying themes for two reasons: childhood anemia is prevalent in the three Asian countries where Creative Health projects are being implemented (first Japan, then Indonesia and Cambodia) [20], and two authors (KEN and SO) are professionals in transfusion medicine and nutrition, respectively. Through the full set of three workshops, students are expected to see relationships emerge between their own health and the community at large. Workshop activities were modified and adjusted through discussions among the research team members and with school teachers.

### 2.2. Field and Workshop Participants

Our Creative Health team engaged with the help of two local retired school principals. Using their network, two schools in Fukushima’s Koori Town and Date City were selected to pilot the three workshops in 2020 and 2021. The Koori Town school implemented all three workshops over the span of a month during each academic year, and the Date City school proceeded likewise in 2021. Student participants were in grade 5 (10–11 years old) in Koori Town and grades 4 to 6 (9–12 years old) in Date City. Our protocol recommends that the workshops target grades 4 to 6, with each school at liberty to select grades depending on their class size and curriculum. Koori Town’s school participated in both years and Date City’s in 2021. Since the principal of the school in Koori Town had attended participatory theater training before [14], this school did the ACT workshop prior to the initiation of our study in the first year. We thus lacked 2020 ACT evaluations from the Koori Town school. In total, 109 students from three schools participated in workshops (Table 1), among whom 3 declined to participate in research and 1 did not submit a survey sheet. Thus, data were analyzed from 105 students, of whom 89% percent were 10 to 11 years old, and a half identified as girls.

### 2.3. Data Items and Analysis

Both quantitative and qualitative data were collected using the same single-page self-administered questionnaire. As for quantitative items, referring to workshop aims, we asked students the following before and after their workshops: “Are you satisfied with your health?” (BODY evaluation indicator), “How do you feel about the food in your place?” (FOOD), and “How you feel about the place you live?” (ACT). To answer each question, students could choose from five face icons: 1, with a neutral expression, through to 5, with the happiest expression. This was deemed to be appropriate for the age range of respondents and the setting. We were giving them an interactive experience, supposedly enjoyable, and courtesy bias in survey responses is common in Japan. Assuming that Japanese students would favor positive ratings, we made the low end of the rating scale start with a neutral face [21]. For basic characteristics, we asked students to self-identify their gender and age. As for qualitative data, at the end of the post-workshop questionnaire, students could add free opinions about how they felt about the workshop and what they learned. We asked class teachers to distribute the questionnaires before the first workshop and collect them right after each workshop.

Stata, version 14.2 (Stata Corp, College Station, TX, USA), was used to analyze quantitative data. We first examined differences in the ratings of three evaluation indicators before and after the workshop, using the Wilcoxon signed-rank test. We then subtracted pre-workshop ratings from post-workshop ratings, and classified students into decreased (lower than zero), no change (zero), and increased (higher than zero) groups. Differences in the proportions of these three groups, depending on study site, year, and self-identified age and gender, were analyzed using the chi-square test.

Text-mining analyses of qualitative data (in Japanese) were conducted using KH Coder, version 3 [22]. Frequently used words were listed, and a sub-graph analysis of a co-occurrence network was conducted in order to classify these words into major topics. For the strength of co-occurrence, Jaccard coefficients were calculated, from which the top 60 (strongest) co-occurrences were drawn in the diagram. A sub-graph analysis (random walks) classified the network into groups that were more closely associated with color-coding. Referring to these results, we conducted correspondence analyses to examine the differences in word usage among the aforementioned three groups (decreased, no change, and increased) and to visualize the results in two dimensions. The relative locations between words and groups show the relative frequencies, as in a contingency table. Words that appear opposite to the contrasting group are characteristic of the target group. The grouping of students who wrote sentences, including such words, was checked one by one, with group-specific words translated into English.

### 2.4. Ethical Consideration

Prior to the first workshop, an informed assent form for students and an informed consent form for parents, along with a cover letter from a school principal, were distributed. Thereafter, signed informed consent forms were collected. The study was approved by the Ethics Committee of Fukushima Medical University (General 2020-060, 16 July 2020).

## 3. Results

### 3.1. Workshop Content

Major activities of the three workshops are presented in Table 2. Each workshop was 90 min in length (with a 10-min break), and applicable to grades 4 to 6. The designer of each workshop was its facilitator in collaboration with the school teachers and project sub-facilitators. During the COVID-19 pandemic, the students’ cooking activity in the FOOD workshop was restricted and replaced by a cooking demonstration and tasting. With international travel restrictions, the second author (ALW), who developed the ACT workshop, could not travel to Japan, so she trained other team members and teachers online (about 2 h) prior to its implementation. Core team members had been involved in previous participatory theater work, and so had some prior training and facilitation experience. Figure 2 shows two scenes of a performance created by students in the ACT workshop. They selected green tea (served at a popular local sushi restaurant) as their favorite drink and performed how tea leaves were picked (left photo), processed, served, and appreciated (right photo). In the FOOD workshop, children drew the meals they had the night before, reviewed nutrition, and added foods to make meals richer in iron. In workshops, facilitators gave instructions on activities, handed out self-study materials, observed, and occasionally provided advice as students worked on their own, and watched their presentations.

Of note, there was a long pause in our field activities in 2020, when the COVID-19 pandemic had just started. While waiting for schools to resume regular teaching routines, two former principals supported a related project with collaborating schools by developing and distributing a COVID-19 prevention leaflet for students and families. This risk communication activity was reported elsewhere [23]. We also created a “Home Adventure” self-directed learning module [24], with seven items that students could try at home to learn various topics and subjects: *Let’s eat*, *Let’s study*, and *Let’s play* (about home life and nutrition); *Let’s exercise* and *Let’s get clean* (applying science and mathematics); *Utilities in your home* and *Good night!* (about safety and what to put into an emergency bag, aspects of social studies, and disaster preparedness). It was an unexpected positive outcome that two schools used the home adventure materials to prepare students for the three workshops that followed. In addition, students and teachers at the school in Koori Town made a 30-min performance of what they had learned from their “Home Adventure” preparatory work and three workshops. This performance was part of an annual school-wide presentation day—attended by parents—in both years.

### 3.2. Student Characteristics and Workshop Evaluation Indicators

For the before-after differences in the students’ evaluation, indicator ratings were statistically significant for BODY (*p* = 0.01) and FOOD (*p* = 0.01). The proportion of students with increased before-to-after ratings was 25% for BODY, 28% for FOOD, and 25% for ACT (Table 3). When differences in the distribution of the three groups (increased, no change, decreased) were analyzed—by study site, year, student’s age, and gender—a proportion of the “increased” group was significantly higher for the Koori Town school in 2021 in the BODY workshop (21% for Date City in 2021, 12% for Koori Town in 2020, and 44% in Koori Town in 2021), and an older age group in the FOOD workshop (14% for 9–10 years old, 41% for 11–12 years old, data not shown). No differences by gender emerged.

### 3.3. Text Analysis

In the BODY evaluation, the total number of sentences written by 105 students was 175. The average count of how many times each word was used in these sentences was 3. The top 5 most frequently used words (limited to nouns) were *blood* (used 33 times), *figure* (27), *heart* (18), *lung* (16), and *pulse* (16). The correspondence analysis (Figure 3) showed distinct differences in words among three groups, and the “no change” group (labeled as 0) was located closest to the origin. The “increased” group (labeled as 1) was characterized by *presentation* and *machine*.

One 11-year-old girl wrote “*I was able to present about Hideyo Noguchi collaboratively (with my classmates)*” (characteristic word underlined). 

Another 11-year-old girl wrote, “*It was good that I was able measure my pulse (81) by a machine*.” 

The “decreased” group (labeled as –1) was characterized by the name of a historical figure, *Noguchi*. 

A 10-year-old boy wrote, “*I learned about Hideyo Noguchi as a historical figure*.”

In the FOOD evaluation, the total number of sentences written by 105 students was 184. The average count of how many times each word was used in these sentences was 4. The top 5 most frequently used words (limited to nouns) were *iron* (used 114 times, of which 77 were written as a kanji ideograph, and 24 were written phonetically), *vitamin C* (77), *vitamin* (20), *foods* (19), and *meals* (12). The “increased” group (labeled as 1) was characterized by *anemia* (Figure 4). 

A 12-year-old girl wrote, “*If there is little iron, anemia occurs. I learned meat is rich in iron and fruits are rich in Vitamin C*.” 

There were no words that were located opposite to the contrasting groups and were clearly characteristic of the “decreased” group (labeled as –1), but *rice* was located farthest from the other two groups. 

A 9-year-old girl wrote, “*I want to cook clam rice*.”

The “no change” group (labeled as 0) was located closest to the origin.

In the ACT evaluation, the total number of sentences written by 61 students was 125. The average count of how many times each word was used in these sentences was 3. The top 5 most frequently used words (limited to nouns) were *Koori* (used 13 times), *food* (9), *game* (8), *body* (8), and *town* (8). The “increased” group (labeled as 1) was characterized by *people* and *learn* (Figure 5). 

A 12-year-old girl wrote, “*Important to collaborate with other people.*” 

Another 11-year-old boy wrote, “*I had very much fun and learned a lot.*” 

There were no words clearly characteristic of the “decreased” group (labeled as –1) but *concentrate* and *body* were located closest to the group label. 

A 12-year-old girl wrote, “*I understand that playing mirror increases concentration.*”

Another 11-year-old girl wrote, “*I think it is important to move your body.*” 

The “no change” group (labeled as 0) was located closest to the origin.

## 4. Discussion

An innovative “Creative Health” project for elementary school children was piloted and assessed in a part of the Fukushima Prefecture affected by a nuclear disaster in 2011, and which is still in the process of recovery. Building on the participatory theater approach (ACT workshop) initiated by a UK researcher, local researchers expanded the project by adding two components (BODY and FOOD workshops). Workshop activities facilitated spontaneous, collaborative work among students that stimulated their scientific and creative thinking about personal health and its connection with the community. Pictures in Figure 1 and Figure 2 show students working as a team to organize their presentation about historical figures in medicine, summarize their opinions about local foods and use performance to explore their favorite local foods. The participatory theater work (ACT) creates a safe space for children to work together to explore issues of food and health in their own environment, encouraging them to take the lead in identifying what is important, what aspects they want to learn more about, and what things could be done differently [25]. The participatory work in the health-related workshops (BODY and FOOD) adopted a flipped-classroom approach [26] that allowed students to learn among themselves by using textual and visual materials supplied by teachers. In both workshops, we further encouraged students to talk with their family members about what they learned [27]. By incorporating such approaches, schools can function as hubs for building community resilience, in which teachers shift from being ‘experts’ to being facilitators who nurture the children’s own knowledge and expertise as well as support them in identifying learning issues that are important for themselves, for families, and for the wider community.

Quantitative before-after evaluations showed that children had more positive perceptions about their health and the foods in their community after attending BODY and FOOD workshops. Although not statistically significant, the same trend was also observed for the ACT workshop, in which students gained more positive views about their community. Our results are supported by recent systematic reviews of studies examining the efficacy of participatory creative activities in the health and well-being of children [28,29]. They included music, dance, singing, drama, and visual arts taking place in community settings or at schools, and found that despite methodological weaknesses, these activities could have positive effects on children’s levels of knowledge, behavioral changes, and, most notably, self-confidence, self-esteem, relationship building, and a sense of belonging. 

The text visualization of free written opinions showed that students whose rating of the workshop evaluation indicators increased had understood and enjoyed the aims and core content of the workshops. Students in the “increased” group appreciated presenting and measuring in the BODY workshop, learned etiology and prevention of anemia in the FOOD workshop, and enjoyed working collaboratively to perform in the ACT workshop. On the contrary, students in the “decreased” group simply reported about the topics they learned (e.g., Hideyo Noguchi in the BODY workshop) and activities they did (e.g., cooking rice in the FOOD workshop and a mirror game in the ACT workshop). 

The MEXT New National Curriculum Standards [19] aims to nurture students’ active and interactive learning skills as reflected in the “increased” group’s opinions. More specifically, the Standards states, “Each school should be committed to enhancing education to enable pupils to solidly acquire basic and fundamental knowledge and skills, to foster pupils’ ability to think, make judgments, and express themselves that are necessary to solve problems using acquired knowledge and skills, to cultivate an attitude of proactive learning to develop pupils’ individuality, and to encourage working together with diverse people”. The set of skills listed in the MEXT Standards (think, make judgments, and express) are included in the skills that students learn in our workshops, as presented in Table 2. The EU CUIDAR project report stated that participatory projects with children and young people must consider them as experts in their own lives, create space for them to express diverse and unique ideas, but be structured enough to guarantee the safety and quality of the project [30]. The balance between creativity and the teaching structure should be discussed further in our project in order to increase the number of students whose ratings of evaluation indicators increase, and who learn ways to learn through working with us. Additionally, there is a need for careful investigation of students whose ratings decrease in order to minimize any potential negative influences of our project. Further development of workshop content and its facilitation should be guided by the MEXT Standards, be discussed with school teachers, and, most importantly, be based on the analyses of students’ responses.

Japanese elementary schools include the following participatory school health activities in their regular curricula: health checkups, school lunch preparation, cleaning activities, evacuation drills, and sports festivals. Tomokawa and colleagues reviewed these activities and identified four key factors for their successful implementation: a legal basis at the national level; an official endorsement of these activities being necessary at prefectural and municipal levels; teacher training and support to carry out activities at the school level; and a shared understanding among teachers, students, and parents in their community [31]. In accordance with these principles, first, the implementation of our Creative Health project is supported by the MEXT New National Curriculum Standards, which promote student active participatory learning. Second, we provided teacher training as reported previously [14] and, in response to the COVID-19 pandemic, we organized online training. Third, with regards to parental involvement, one school assigned our “Home Adventure” to students as independent study and invited parents to view the students performing what they had learned through the workshops. This indicated that our project offered a creative space for teachers to adopt and adapt our Creative Health teaching materials in their own way.

Three major methodological limitations pertain to the present study. First, the before-after comparison design lacked a control group. As addressed by Phillips [32], a more rigorous formative assessment of the creative health works, along with a comprehensive review of existing evidence, is needed. A second issue is the small sample size, as suggested by a change in student ratings about the ACT workshop not reaching statistical significance. Although we observed a few differences in student responses depending on the study site and age, the modification of workshop content—depending on the school and student characteristics—will require further analyses by adding data from future workshops. The third is our assessments of the teachers’ and students’ responses being reported separately in a previous publication [14] and in the present study. A triangulation of students’ and teachers’ opinions are needed for a more comprehensive understanding of both the advantages and limitations of the project in order to improve it.

## 5. Conclusions

More than a decade has passed since the 2011 Fukushima nuclear disaster, during which it has become more evident that community restoration requires a long-term vision of how to repair interwoven social issues [33]. In present-day Fukushima, children and young adults who grew up in the aftermath of the disaster, younger children who did not experience the disaster, and adults who have observed many changes are living together. This provides a meaningful context for children and adults to work together for a participatory system that builds community resilience. The three workshops in our Creative Health project facilitated students to work collaboratively, think scientifically and creatively, and express their opinions. Workshop participation resulted in students having more positive perceptions about their health, the foods in their community, and their community itself. Teaching tools made by project facilitators were put to good use by teachers in collaboration with parents for independent study. The Creative Health project could be a way for teachers to work with the next generation by nurturing children’s creativity and autonomy in expressing issues of concern to them, enhancing children’s capacities as agents of change in their community, and linking schools with the communities they serve.

## Figures and Tables

**Figure 1 ijerph-19-03417-f001:**
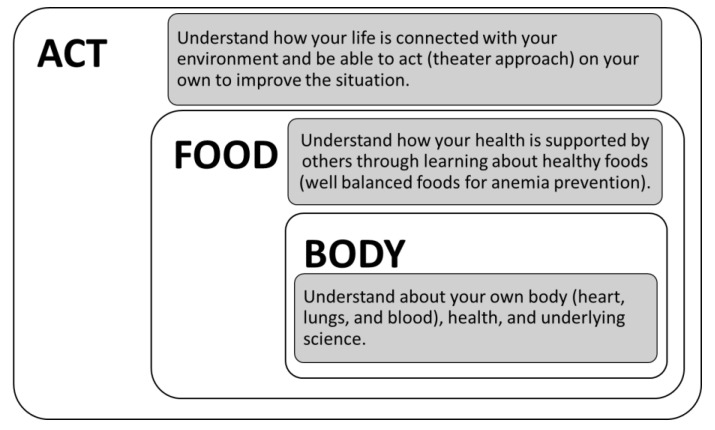
Aims of three Creative Health workshops. Students discuss their favorite foods in the ACT workshop and how to introduce historical figures in medicine during the BODY workshop. Details of the workshops are presented in the following tables.

**Figure 2 ijerph-19-03417-f002:**
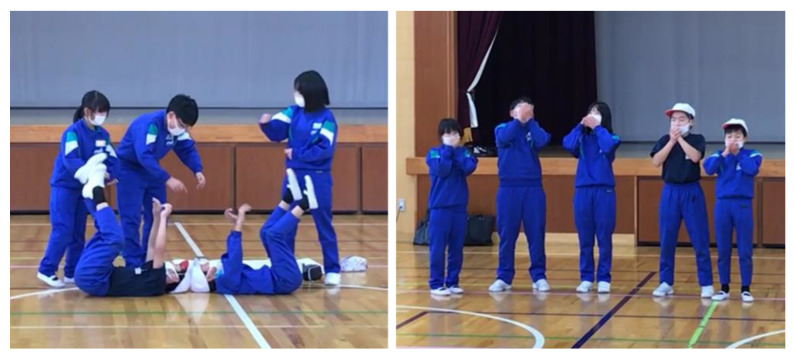
Students’ performance in the ACT workshop. Students’ performance “green tea” in the ACT workshop. They performed how tea leaves were picked (**left**), processed, served, and appreciated (**right**).

**Figure 3 ijerph-19-03417-f003:**
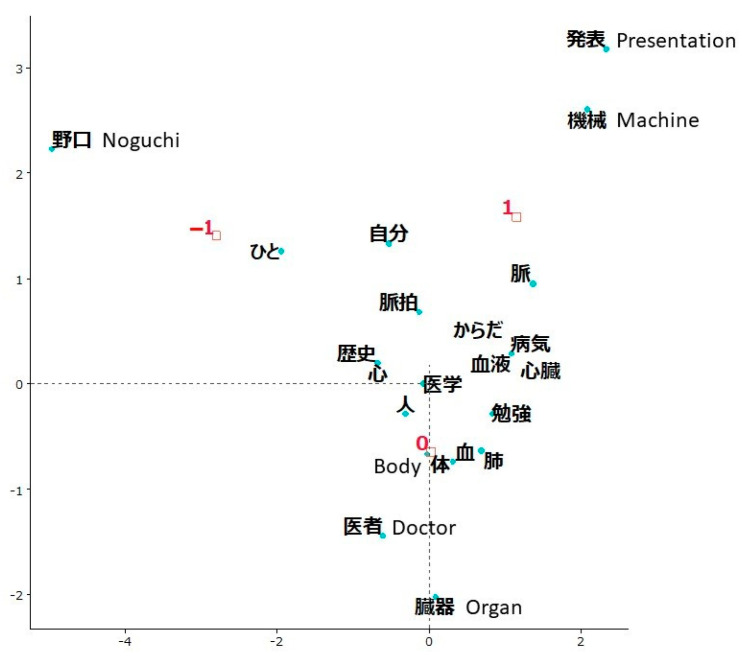
Correspondence analysis of words used in students’ opinions about the BODY workshop depending on changes in the evaluation indicator rating. The group with an increased rating of the workshop evaluation indicator was labeled as 1, no change in rating as 0, and a decreased rating as –1. Words that were not characteristic of any group were *person, me, pulse, body, disease, history, blood, heart, medicine, study*, and *lungs*.

**Figure 4 ijerph-19-03417-f004:**
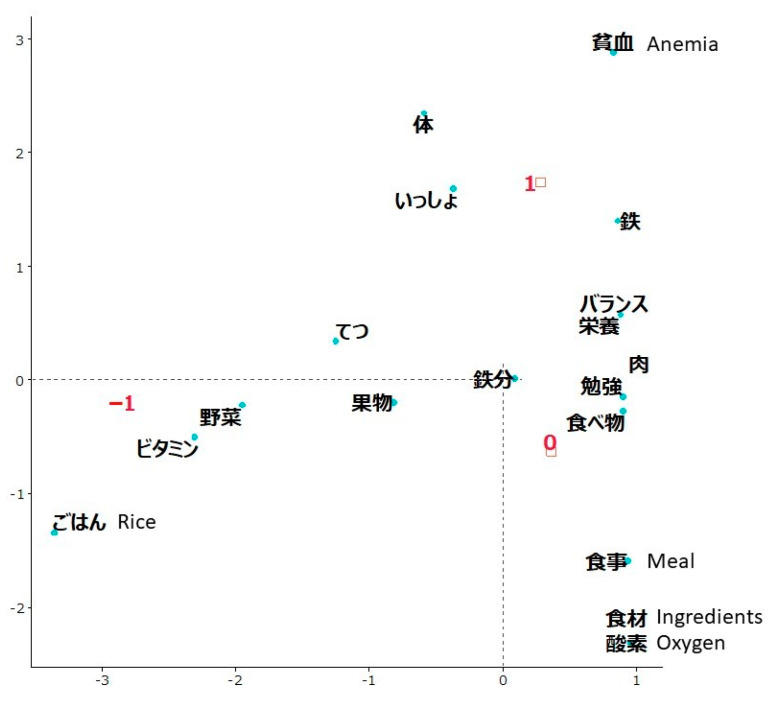
Correspondence analysis of words used in students’ opinions about the FOOD workshop depending on changes in the evaluation indicator rating. The group with an increased rating of the workshop evaluation indicator was labeled as 1, no change in rating as 0, and a decreased rating as –1. Words that were not characteristic of any group were *body*, *together*, *iron*, *balance*, *nutrition*, *iron*, *meat*, *study*, *food*, *fruit, vegetable*, and *vitamin*.

**Figure 5 ijerph-19-03417-f005:**
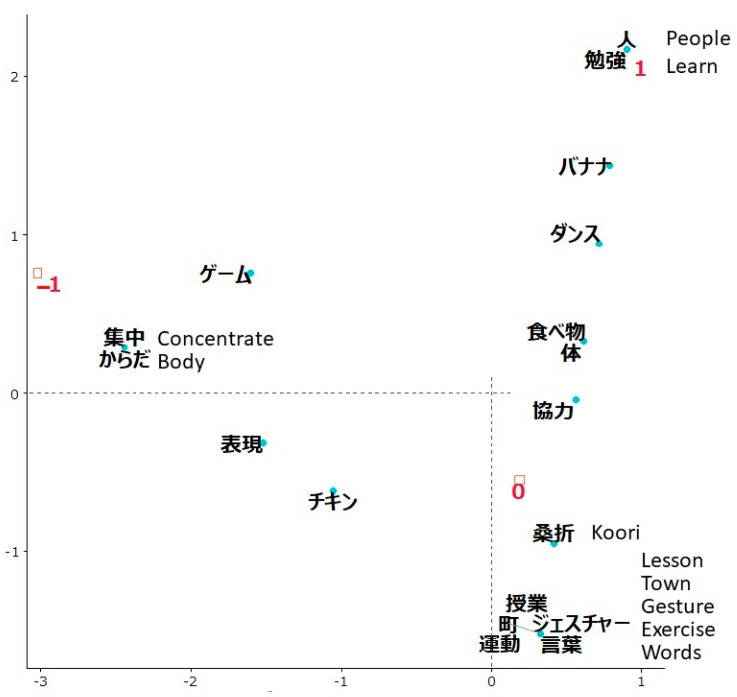
Correspondence analysis of words used in students’ opinions about the ACT workshop depending on changes in the evaluation indicator rating. The group with an increased rating of the workshop evaluation indicator was labeled as 1, no change in rating as 0, and a decreased rating as –1. Words that were not characteristic of any group were *Banana* (name of a song)*, dance, game, food, body, collaborate, express,* and *Chicken* (name of a game).

**Table 1 ijerph-19-03417-t001:** Characteristics of participating students.

	*n* (%) Total *n* = 105
Site and year	
Koori Town, 2020	44 (41.9)
Koori Town, 2021	36 (34.3)
Date City, 2021	25 (23.8)
Age	
9 years old	4 (3.8)
10 years old	43 (41.0)
11 years old	50 (47.6)
12 years old	5 (4.8)
Missing	3 (2.9)
Gender	
Boy	46 (43.8)
Girl	52 (49.5)
Not wanting to answer or missing	7 (6.7)

**Table 2 ijerph-19-03417-t002:** Content of three Creative Health workshops.

Workshop Activities	Required Skills
**BODY**	
1. Introduction (5 min) Seat in groups of 5–6 people Self-introduction of the main facilitator using storyboards Explain workshop aims	
2. Individual work (20 min) Write on sticky notes some knowledge, belief, or experience related to blood, heart, and/or lungs Post sticky notes on a Venn diagram (overlapping circles marked as blood, heart, and lungs) Class teachers or sub-facilitators comment on what they learned from the notes	Logical thinking Sharing ideas
3. Teamwork 1 (30 min with 10 min break) Student groups receive and examine storyboards. Each group is assigned to a historical figure Plan each person’s role (e.g., playing the historical figure, narrating the historical figure’s contributions to medical science, showing storyboards in a way that everyone can see), and practice Present to the whole class	Work collaboratively Public speaking
3. Teamwork 2 (15 min) Students in each group share a pulse oximeter to measure and record their heart rates Plot heart rates on a Cartesian grid and study its shape	Scientific and mathematical thinking
4. Wrap up (10 min) Facilitator(s) comment on student accomplishments Students take turns mentioning things they learned, enjoyed, and plan to do from now on	
**FOOD**	
1. Introduction (10 min) Seat in 4–5 groups Explain workshop aims Demonstration of iron-rich cooking (e.g., clam rice)	
2. Main activity 1 (25 min) Learn about roles of iron in body from car cartoons (car seats represent iron) Learn to pick foods rich in iron and vitamins from samples and pictures (including local foods)	What they eat connects with their health
4. Main activity 2 (25 min with 10 min break) Draw what they ate for dinner the day before Discuss what to add to make it more iron-rich Present the iron-rich menu to others	Review daily eating from drawings How foods are prepared by their families and in community
5. Tasting (10 min)	
6. Wrap up (10 min) Facilitator(s) comment on student accomplishments Invite students to ask any questions they want to ask	
**ACT**	
1. Introduction (5 min) Gather in a circle Greetings and introductions Explain workshop aims	
2. Warm-up (10 min) Active exercise: “Chicken” (counting with actions) or other active warm-ups Concentration exercise: “Mirrors” (mirror actions of the other in a pair)	Express with actions and voice Work collaboratively
3. Main activity 1 (20 min) Make groups of 5–6 people Think about favorite foods in community and discuss how they are produced, prepared, and eaten. Pick one food and present reasons for choice	Think and discuss about foods and how they are produced, prepared, and eaten in their community Making choices for making a performance
4. Main activity 2 (35 min with 10 min break) Create a short drama scene about the food (about 30 s) using actions and sounds Practice and rehearse Show back to each other Facilitator(s) and students comment on each performance	Create a performance to explore and express ideas to others
5. Review and closing (10 min) Gather in circle Invite students to discuss in pairs what they enjoyed, what they learned and any questions arising from today’s workshop. Share reflections with the whole group and discuss anything they want to follow up or learn more about. Finish with “Chicken” (or other activity)	

**Table 3 ijerph-19-03417-t003:** Changes in workshop evaluation indicators.

	*n* (%) ^a^			
Indicators	Pre-Workshop	Post-Workshop	*n* (%) ^a^	*p*-Value ^b^
BODY“ Are you satisfied with your health?”				
Rating ^c^				
1	10 (9.7)	7 (7.1)		0.01
2	14 (13.6)	8 (8.1)		
3	39 (37.9)	33 (33.3)		
4	19 (18.5)	30 (30.3)		
5	21 (20.4)	21 (21.2)		
Post-workshop minus pre-workshop rating				
Decreased			10 (10.2)	
No change			64 (65.3)	
Increased			24 (24.5)	
FOOD “How do you feel about the food in your place?”				
Rating ^c^				
1	5 (4.9)	5 (5.3)		0.01
2	14 (13.7)	3 (3.2)		
3	17 (16.7)	19 (20.0)		
4	31 (30.4)	28 (29.5)		
5	35 (34.3)	40 (42.1)		
Post-workshop minus pre-workshop rating				
Decreased			11 (11.7)	
No change			57 (60.6)	
Increased			26 (27.7)	
ACT “How do you feel about the place you live?”				
Rating ^c,d^				
1	9 (8.8)	2 (3.3)		0.13
2	13 (12.8)	10 (16.4)		
3	22 (21.6)	10 (16.4)		
4	27 (26.5)	15 (24.6)		
5	31 (30.4)	24 (39.3)		
Post-workshop minus pre-workshop rating				
Decreased			8 (13.6)	
No change			36 (61.0)	
Increased			15 (25.4)	

^a^. Due to missing data, some totals do not add up to 105. ^b^. Wilcoxon sign-rank test was used. ^c^. Five-level face scale was used, with 1 corresponding to a neutral expression and 5 to a large smile. ^d^. Evaluation of Koori Town school in 2020 was not included since the school conducted this workshop prior to start of our research.

## Data Availability

The data are not publicly available due to the nature of this research that students and parents did not agree for their data to be publicly shared.

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
