# Peer review of "Empowering Children as Agents of Change to Foster Resilience in Community: Implementing “Creative Health” in Primary Schools after the Fukushima Nuclear Disaster"

_ijerph, 2022, doi:10.3390/ijerph19063417_

Round 1
Reviewer 1 Report
Dear Authors,
I have very much enjoyed reviewing your manuscript. It offers a very creative and important response to the challenges experienced in the aftermath of a disaster of the magnitude Japan witnessed in 2011. It reminds us of the value and need for engaging with the arts at a time when so much of society only values the sciences. To be fully human, we need both. To recover from tragedy, fear, stress and all of the deep trauma that comes with living our lives, the arts play an even more significant role. For this reason, you paper has even more currency in today's climate.
Reviewer 2 Report
Comments and suggestions:
- About the title: The title is not so linked with the keywords. The word "actors" in the title is a little bit confusing because the meaning is not clear at first reading. If the Creative Health program pretends to foster community resilience (and it is a keyword), would be interesting to be shown in the title to understand the study. For example: Empowering children as actors in community: Implementing “Cre-2 ative Health o foster resilience in primary schools after the Fukushima nuclear disaster.
- In the abstract: The aim of the study is understood, but is not so clear. Should be more specified. Also, in the manuscript, it is clear the aim of "Creative Health", but the aim of the study/article should be specified.
- In the introduction: It lacks a description and references about what is understood for being "actors of the community".
- In all the parts of the paper: Sometimes the Creative Health is called project and sometimes program. If it is a program, it's important to be always called program for not confusing the reader.
- About the references: are relevant and current.
- About 2.1: The program and the development is well described, and the pictures help to imagine the procedure.
- About the participants (2.2.), ¿how many students, in total, participated in the study? The table 2 is shown as a result, but it should be presented in 2.2. to describe the characteristics of the participants.
- About the field (2.2), how many time did the implementation of the workshop take in both years? Can you facilitate a schedule?
- Would be interesting to divide the field and the participants in 2.2. and give more details for each one.
- About data items (2.3): It is understood that the same tool collected both quantitative and qualitative data. It is suggested to include a table to describe which variables are included in quantitative questions and which variables are included in quantitative questions of the same questionnaire.
- Table 1 should be in the section 2.1. because describe the characteristics of the program, it is not a result.
- Table 2 should be in the section 2.2. because describe the characteristics of the participants, it is not a result.
- 3.1. is procedures, should be in 2.2. as Field.
- 3.2. is description of the participants, should be in 2.2. as Participants.
- The real results start in 3.3. The results should describe clearly the qualitative data, with "quotes" of the participants. At the same time, could be interesting that the qualitative data would be shown linked with quantitative data.
- The discussion is correct and the results of the study are compared with relevant references.
- The limitations of the study are clear.
- The conclusions are important to the knowledge field.
Reviewer 3 Report
The school children's views and understanding of crisis and disaster management is important to all educational and social systems and that makes the topic of the paper of significance.
The method and findings are clearly presented. However, authors create high expectations of findings regarding arts-based creative thinking and action that are not met. Although the study is inspired by the All-Party parliamentary Report 'Creative Health: the Arts for Health and Wellbeing' (2017), there are very minimum indicators of creative and artistic activities being used in the workshops serving the study. I would have hoped to read about the use of applied drama/theatre and participatory activities (Augusto Boal's Forum Theatre for example) towards engaging school children in developing critical thinking about bodies, food and behaviour. I would be interested to read more around somatic awareness, dance and movement as tools of knowing the body and reflecting on healthy eating attitudes. The study is very much education-focused and although it claims intentions of creative and critical thinking amongst child populations, it remains strictly patronised, in my view, as it is limited by traditional learning tools.
It would have been helpful if authors could
(a) provide us with their own definition of creativity in a Japanese educational and cultural context and,
(b) explain how the arts have been used with respect to the artistic process (not just as learning tools) towards enabling interrogation and learning through interaction and experimentation. Give specific examples of using arts forms in workshops (how and why).
The project, as it is presented, does not meet the request of using the creative arts in disaster and crisis management.
